# Homozygous AA Genotype of IL-17A and 14-bp Insertion Polymorphism in HLA-G 3′UTR Are Associated with Increased Risk of Gestational Diabetes Mellitus

**DOI:** 10.3390/ijerph22030327

**Published:** 2025-02-22

**Authors:** Amaxsell Thiago Barros de Souza, Cecília Rodrigues Lucas, Kleyton Thiago Costa de Carvalho, Antonia Pereira Rosa Neta, Emanuelly Bernardes-Oliveira, Juliana Dantas de Araújo Santos Camargo, André Ducati Luchessi, Ricardo Ney Cobucci, Janaina Cristiana de Oliveira Crispim

**Affiliations:** 1Postgraduate Program in Sciences Applied to Women’s Health, Federal University of Rio Grande do Norte, Natal 59012-310, Brazil; thiagoamaxsell@gmail.com (A.T.B.d.S.); ricardo.cobucci.737@ufrn.edu.br (R.N.C.); 2Faculty of Pharmacy, Federal University of Rio Grande do Norte, Natal 59012-570, Brazil; cecilia.rlucas@gmail.com; 3Department of Clinical and Toxicological Analysis, Federal University of Rio Grande do Norte, Natal 59012-570, Brazil; kleyton.carvalho@ufrn.br (K.T.C.d.C.); andre.luchessi@ufrn.br (A.D.L.); 4Postgraduate Program in Health Sciences, Federal University of Rio Grande do Norte, Natal 59012-570, Brazil; antoniaprosaneta@gmail.com (A.P.R.N.); juliana_ily@hotmail.com (J.D.d.A.S.C.); 5Postgraduate Program in Development and Technological Innovation in Medicines, Federal University of Rio Grande do Norte, Natal 59012-570, Brazil; bio_natalrn@yahoo.com.br; 6Januário Cicco Maternity School, Brazilian Company of Hospital Services (EBSERH), Natal 59012-310, Brazil

**Keywords:** Gestational Diabetes Mellitus, *IL-17A* polymorphism, *IL-17RA* polymorphism, HLA-G 14-bp insertion/deletion, genetic association

## Abstract

Gestational diabetes mellitus (GDM) is a common pregnancy complication characterized by hyperglycemia and insulin resistance, with unclear genetic mechanisms. The specific involvement of proinflammatory cytokines, including IL-17A, and the immuno-tolerogenic HLA-G remains poorly understood in GDM. We aimed to explore the associations of three polymorphisms, *IL-17A* -197G>A (rs2275913), *IL-17RA* -947A>G (rs4819554), and HLA-G 14-bp insertion/deletion (indel), with GDM risk in a Brazilian population. We conducted a case-control study (79 GDM cases and 79 controls). Genetic polymorphisms were analyzed using PCR–RFLP, with DNA extracted using the Salting-out procedure. Significant associations were identified between -197G>A rs2275913 and HLA-G 14-bp indel polymorphisms in both codominant and recessive models. The *IL-17A* rs2275913 AA genotype was associated with a nearly ten-fold increased risk of GDM in both the codominant (*p* = 0.021, OR 9.89, 95% CI: 1.63–59.92) and recessive models (*p* = 0.006, OR 9.33, 95% CI: 1.57–55.38). Similarly, the HLA-G 14-bp Ins/Ins genotype was associated with an increased risk in both the codominant (*p* = 0.026, OR 3.34, 95% CI: 0.98–11.41) and recessive models (*p* = 0.010, OR 4.20, 95% CI: 1.36–12.96). *IL-17RA* polymorphism showed no significant associations. The study findings highlight the potential genetic and immune factors associated with GDM, particularly the -197G>A rs2275913 and HLA-G 14-bp indel polymorphisms. Further functional characterization is warranted to uncover the mechanism of genotype–phenotype association.

## 1. Introduction

Research on maternal health, particularly in relation to Gestational Diabetes Mellitus (GDM), has intensified owing to the significant risks and serious adverse health outcomes associated with this condition. Various metabolic changes occur during pregnancy, and carbohydrate intolerance can lead to hyperglycemia and insulin resistance, thereby increasing the risk of developing GDM [1]. Globally, GDM is the most common medical complication during pregnancy, affecting up to 25% of pregnant women [2]. In Brazil, the prevalence ranges from 14% to 15% [3]. However, the primary causes and pathogenic mechanisms underlying GDM remain poorly understood [4].

It has been reported that both proinflammatory cytokines and checkpoint molecules, such as human leukocyte antigen (HLA)-G, are essential for the development and continuation of pregnancy from its inception to completion [4,5,6,7]. Studies have shown that GDM is characterized by an imbalance favoring pro-inflammation over the anti-inflammatory system when compared to non-pregnant women [8,9]. This imbalance highlights the need for detailed explanation and follow-up of GDM. In addition to genetic and immunological factors, recent studies have indicated that clinical and anthropometric markers may improve the prediction of GDM. For example, the integration of abdominal subcutaneous fat thickness (ASFT) with the 50-g glucose challenge test (GCT) has been proposed as a potential predictive tool for GDM, emphasizing the multifactorial nature of this condition and the necessity for integrated diagnostic approaches [10]. Moreover, the predictive value of inflammatory markers, such as C-reactive protein (CRP) and interleukin-6 (IL-6), has been demonstrated to correlate with GDM risk, thereby underscoring the significance of systemic inflammation in its pathogenesis [11].

Evidence suggests that the fetal antigen load might trigger events leading to GDM. In this context, *HLA-G*, an anti-inflammatory class Ib gene located within the major histocompatibility complex (MHC) at 6p21.3 [12,13], has garnered attention. HLA-G is known to stimulate the production of angiogenic factors and cytokines that promote embryo implantation, placental vascularization, and maternal–fetal tolerance [14]. The immunomodulatory role of HLA-G is crucial for maintaining immune homeostasis during pregnancy, as it prevents excessive maternal immune responses against the semi-allogeneic fetus. Dysregulation of HLA-G expression has been associated with pregnancy complications, including preeclampsia and recurrent pregnancy loss [15,16]. A polymorphic site in the 3′ untranslated region (UTR) of the *HLA-G* gene, characterized by an insertion or deletion of 14-base pairs (bp), influences mRNA stability and, subsequently, HLA-G protein expression [17]. Low levels of soluble HLA-G (sHLA-G) in the blood have been associated with immune activation at the maternal–fetal interface, potentially leading to adverse pregnancy outcomes, including GDM [18,19,20,21,22]. Paradoxically, one study identified an association between the *HLA-G* Del/Del genotype, elevated sHLA-G concentrations, and GDM in Italian women [23]. These findings suggest an inconsistency in the role of HLA-G and its insertion/deletion (indel) polymorphism in the pathogenesis of GDM.

Additionally, pregnancy is a pro-inflammatory state, with normal pregnancies exhibiting low-grade inflammation. A delicate balance between pro- and anti-inflammatory cytokines is essential for maintaining a healthy pregnancy [24]. Proinflammatory cytokines, such as interleukin 1β (IL-1β), tumor necrosis factor α (TNF-α), IL-6, and interleukin 17 (IL-17), play significant roles in the pathophysiology of GDM [25,26]. Although polymorphisms in the IL-17 gene family have been implicated in GDM [27], the association between specific single nucleotide polymorphisms (SNPs) in the promoter regions of the *IL-17A* -197G>A (rs2275913) and *IL-17RA* -947A>G (rs4819554) genes and GDM remains unexplored in Brazilian populations and requires further investigation. Given the population-specific genetic variations in cytokine regulation and immune tolerance, further studies are necessary to elucidate how these variants interact with the environmental and epigenetic factors that influence GDM susceptibility.

Molecularly, these SNPs can affect the balance between pro- and anti-inflammatory signaling. This immune dysregulation may contribute to systemic inflammation and insulin resistance, further highlighting the relevance of these polymorphisms in GDM pathophysiology. The -197G>A rs2275913 and -947A>G rs4819554 polymorphisms may increase proinflammatory cytokine production, driving an imbalance in favor of inflammation, which is a characteristic feature of GDM [25,26]. In contrast, *HLA-G* 14-bp indel polymorphism influences the immune response by modulating its soluble protein [17]. This interplay between the pro- and anti-inflammatory pathways at the molecular level is crucial for understanding how immune dysregulation contributes to GDM development.

Therefore, this study aimed to investigate the potential involvement of -197G>A rs2275913, -947A>G rs4819554, and *HLA-G* 14-bp indel polymorphisms in the genetic predisposition to GDM. These findings may offer valuable insights into the early diagnosis and management of GDM.

## 2. Materials and Methods

### 2.1. Study Population and Design

This case-control study, conducted between May 2018 and December 2019, enrolled a total of 79 pregnant Brazilian women clinically diagnosed with GDM and 79 healthy pregnant controls without comorbidities at a teaching hospital in northeastern Brazil. GDM diagnosis was based on results of the 75 g 2-h oral glucose tolerance test (OGTT) performed at the first prenatal visit or between 24 and 28 weeks of pregnancy, following the guidelines of the American Diabetes Association (ADA) [28].

Exclusion criteria included a history of autoimmune diseases, hypertension, prior diabetes mellitus diagnosis (fasting glucose: 96–120 mg/dL), infections, and age < 18 years. Written informed consent was obtained from all the participants for the use of peripheral blood samples for genotyping. The study protocol was approved by the institutional ethics committee (protocol number 2.631.092 on 2 May 2018, CAAE: 73305717.2.0000.5292).

### 2.2. Samples and DNA Extraction

Genomic DNA (gDNA) was extracted from peripheral blood using the Salting-out method described by Salazar et al. (1998) [29], with minor modifications. DNA quality and integrity were evaluated by 1% agarose gel electrophoresis, and quantification was performed using a Nanodrop spectrophotometer. Amplification of the β-globin gene served as an internal control to confirm the DNA integrity.

### 2.3. HLA-G 14pb Gene Polymorphisms

The *HLA-G* 14-bp indel polymorphism in the 3′UTR of exon 8 (rs66554220) was analyzed. A total of 200 ng of gDNA was amplified in a 25 μL reaction mixture containing 1.0 μM each of forward and reverse primers, 1.5 mM MgCl_2_, 0.2 mM dNTPs, 1 U Taq DNA polymerase, and 100 ng DNA template. Genotyping was performed using polymerase chain reaction-restriction fragment length polymorphism (PCR–RFLP), followed by 3.5% agarose gel electrophoresis, along with a standard DNA ladder. Fragment sizes were determined by comparison with a 100-bp molecular weight marker. The presence of a 345-bp fragment indicated the deletion allele, while a 359-bp fragment represented the insertion allele. The quality control was performed by re-genotyping 10% of the samples. The primer sequences and PCR protocols are detailed in Table 1.

### 2.4. IL-17A/RA Gene Polymorphisms

The genotypes of *IL-17A* (rs2275913; -197G>A) and *IL-17RA* (rs4819554; -947A>G) SNPs were determined using PCR–RFLP. Amplification was performed using specific primers in a 25 μL reaction volume containing 0.5 μM each of forward and reverse primers, 1.5 mM MgCl2, 0.2 mM dNTPs, 1 U Taq DNA polymerase, and 100 ng DNA template. The reactions were carried out in a B960 Advance (Even) thermal cycler. Five microliters of PCR product were digested with XagI (Promega, Madison, WI, USA) for IL-17A and PvuII (Thermo Scientific, Waltham, MA, USA) for IL-17RA. The resulting fragments were separated on 2% agarose gels (Uniscience, São Paulo, SP, Brazil), along with a standard DNA ladder, and visualized to determine their genotypes. Primer sequences and PCR protocols are detailed in Table 1.

To minimize contamination, all PCR procedures were conducted in a designated area with distinct workspaces for DNA extraction, reaction setup, amplification, and post-PCR analysis. Aerosol-resistant tips and DNA-free reagents were utilized to prevent cross-contamination. Negative controls (reactions devoid of DNA template) were incorporated into each PCR–RFLP reaction. Furthermore, pipettes were regularly decontaminated, and reagent aliquots were prepared in small volumes to mitigate the effects of repeated freeze–thaw–cycles.

### 2.5. Statistical Analysis

The Shapiro–Wilk test was employed to assess the distribution of continuous variables in both the case and control groups. Variables exhibiting deviation from normality are presented as medians with interquartile ranges (25th and 75th percentiles), whereas qualitative variables are summarized as counts and percentages. The Mann–Whitney U test was utilized to compare non-normally distributed variables between groups. Associations between groups and categorical variables were analyzed by employing the Chi-square test, and Fisher’s exact test was applied in instances where the expected frequencies in table cells were less than five. All analyses were conducted using R version 4.4.1 within RStudio version 2024.09.1+394 [30].

The Hardy–Weinberg equilibrium (HWE) was assessed for each SNP using the SNPassoc package in R version 4.4.1. This method employs an exact test to determine whether the observed genotype frequencies in the control group deviate from the expected frequencies under HWE. SNPs in the control group with a *p*-value < 0.05 were considered to deviate from equilibrium and were excluded from further analysis. The association between case-control status and SNP effects was evaluated utilizing the SNPAssoc package in R [31], testing three inheritance models: codominant, dominant, and recessive. The analyses were repeated with adjustments for covariates, including age, fasting blood glucose (categorized as >95 mg/dL or ≤95 mg/dL), and family history of diabetes. Exclusions (one control and one case) occurred due to missing data for at least one of these covariates, precluding their inclusion in the analysis. Statistical significance was established as a two-tailed *p*-value < 0.05, with Benjamini–Hochberg False Discovery Rate (FDR) adjustments for multiple comparisons. A power analysis was performed using the pwr package in R to determine the appropriate sample size, setting α = 0.05, 80% of power, and a medium effect size (w = 0.3) [32]. The analysis showed that the sample size used in this study exceeds the minimum required to achieve sufficient statistical power for detecting meaningful effects.

## 3. Results

### 3.1. Clinical Data

This study included 158 women (79 cases and 79 controls); the median age of patients in the GDM group was 31 years, while that in the control group was 30 years. No significant differences were observed in age, gestational age, family history, overweight status, or abortion rate. The demographic and clinical characteristics of patients are summarized in Table 2.

### 3.2. Genotype and Allelic Frequency

The HWE values in Table 3 correspond to both the case and control groups. In the control group, all values were HWE > 0.05 (*IL-17A* = 1.0, *IL-17RA* = 0.0595, *HLA-G* 14-bp = 0.1543) (Table 3).

Table 3 lists allelic and genotype frequencies. The distribution of the -197G>A rs2275913, -947A>G rs4819554, and *HLA-G* 14-bp indel was analyzed in women with GDM and controls. For the -197G>A rs2275913 polymorphism, the allele frequencies were 24.1% (A) and 75.9% (G) in the GDM group and 15.4% (A) and 84.6% (G) in the control group. The genotype distribution was 63.3% (G/G), 25.3% (G/A), and 11.4% (A/A) in the GDM group and 71.6% (G/G), 25.9% (G/A), and 2.5% (A/A) in the controls.

For the -947A>G rs4819554, allele frequencies were 31.6% (G) and 68.4% (A) in the GDM group and 28.4% (G) and 71.6% (A) in the control group. The genotype distribution was 49.4% (A/A), 38.0% (A/G), and 12.7% (G/G) in the GDM group compared to 55.6% (A/A), 32.1% (A/G), and 12.3% (G/G) in the controls.

Regarding the *HLA-G* 14-bp indel polymorphism, the Del allele was found in 58.9% of the GDM group and 62.3% of the control group, whereas the Ins allele was present in 41.1% of the GDM group and 37.7% of the control group. The genotype distribution for this SNP was 19.0% (Ins/Ins), 44.3% (Ins/Del), and 36.7% (Del/Del) in the GDM group, and 9.9% (Ins/Ins), 55.6% (Ins/Del), and 34.6% (Del/Del) in the control group.

### 3.3. Distribution Models

Only the recessive model of *IL-17A* -197G>A rs2275913 was statistically significant. According to these results, IL-17A may be associated with the risk of GDM, with the risk being nearly five-fold times higher in individuals with AA genotype (*p* = 0.023; OR 4.95, 95% CI 1.03–23.69) (Table 4).

As presented in Table 5, the analysis was adjusted for covariates, including age, fasting blood glucose levels (categorized as >95 mg/dL or ≤95 mg/dL), and family history. Specifically, for the *HLA-G* 14-bp indel polymorphism, both the codominant and recessive models indicate that individuals with the Ins/Ins genotype exhibit higher odds ratios (OR) of 3.34 and 4.20, respectively. Similarly, for *IL-17A*, both the codominant and recessive models demonstrate that individuals with the AA genotype have elevated ORs of 9.89 and 9.33, respectively. We tested whether the risk of *IL-17A* for GDM under the recessive model differed among patients with or without a history of GDM, abortion, family history, being overweight, or hyperglycemia. However, no significant association was observed between these interactions. Figure 1 illustrates the proposed mechanism of action of the *IL-17A* -197G>A (rs2275913) and *HLA-G* 3′UTR 14-bp indel SNPs in GDM.

## 4. Discussion

Several genome-wide association studies (GWAS) have identified numerous susceptibility genes involved in the pathogenesis of complex diseases [33,34]. Among these, studies have highlighted the association of IL-17 SNPs with diabetes, including both autoimmune type 1 (T1DM) and type 2 diabetes mellitus (T2DM) [35,36,37,38]. In this study, we evaluated the distribution of SNPs in *IL-17A* -197G>A rs2275913, *IL-17RA* -947A>G rs4819554, and the *HLA-G* 14-bp indel to determine their involvement in GDM. Our findings revealed a significant association between the homozygous AA codominant and recessive genotypes of *IL-17A* and GDM, as well as between the Ins/Ins genotype of *HLA-G* 14-bp in a codominant and recessive model.

Few studies have investigated the role of proinflammatory cytokines, including *IL-17A*, in GDM pathogenesis [39]. Evidence also suggests that IL-17 is responsible for various immune responses, including the mechanisms underlying embryo rejection, which may contribute to abortions and complications during the gestational period [40]. Furthermore, the role of polymorphic loci in the regulation of cytokine expression in GDM has been insufficiently explored. The *IL-17A* -197G>A rs2275913 SNP has been proposed as a critical modulator of gene transcription, with evidence suggesting that the A allele exhibits strong affinity for the nuclear factor of activated T cells (NFAT) [41,42,43,44]. Within T cells, NFAT orchestrates regulatory activity and influences proliferation and differentiation. In conjunction with ROR-γt, activator protein 1 (AP-1), signal transducer and activator of transcription 3 (STAT3), aryl hydrocarbon receptor (AhR), interferon regulatory factor 4 (IRF4), and Runx1, NFAT binds to the IL-17 promoter region, thereby stimulating gene transcription [43]. Liu et al. concluded that the presence of the A allele is associated with the upregulation of IL-17A owing to increased affinity for NFAT [44]. Similarly, a study showed higher serum levels of IL-17 associated with A allele in rs2275913 SNP [45].

Previously, studies showed an association between *IL-17A* -197G>A rs2275913 SNP and several clinical conditions [45,46,47,48]. Evidence also supports that the A allele is more prevalent in patients with T2DM and has been associated with autoimmune T1DM [36,37]. A recent study found that, while the genotype frequencies of the *IL-17A* +45G>A rs3819025 SNP were significantly different between GDM patients and controls, with a predominance of the A allele, the -197G>A rs2275913 SNP showed no significant difference [27]. In our study, for the first time, the homozygous AA codominant and recessive genotypes of the *IL-17A* -197G>A rs2275913 SNP were associated with GDM. Therefore, our results support the hypothesis that the A allele acts as a probable positive regulator of *IL-17A* transcription and strongly contributes to the pathogenesis of GDM.

IL-17 is implicated in insulin resistance through multiple mechanisms, including the activation of inflammatory cells, such as macrophages, which subsequently secrete pro-inflammatory cytokines, such as TNF-α and IL-6 [49]. These cytokines disrupt normal insulin signaling in target cells [50]. Moreover, IL-17 can directly interfere with the insulin signaling pathway, particularly by acting on differentiated adipocytes, leading to impaired cellular responsiveness to insulin and reduced glucose uptake [51].

Additionally, Li et al. found that the GG genotype of SNP -947A>G rs4819554 in the *IL-17RA* gene loci was associated with a decreased risk of autoimmune T1DM [35]. Significantly higher levels of HbA1c have also been reported in T1DM carriers of the genotype with the A allele [37]. Increased IL-17 and HbA1c levels are associated with oxidative stress and insulin resistance [52,53]. Insulin resistance causes an increase in reactive oxygen species, which increases HbA1c levels. Oxidative damage leads to an increase in proinflammatory cytokines, resulting in elevated IL-17A levels [52,53]. However, our results showed no significant association between the different genetic models of the -947A>G rs4819554 SNP and GDM. The G allele leads to lower IL-17RA expression because of its reduced binding affinity to polymerase during transcription. This decrease in expression results in reduced IL-17A [54,55]. Interestingly, we observed a higher frequency of the A allele (68.4%) in women with GDM. We propose that there is a significant increase in the expression of IL-17RA, which supports the hypothesis that elevated IL-17A pro-inflammatory activity plays a significant role in the pathogenesis of GDM.

In contrast, HLA-G is critical in maternal–fetal immune tolerance, as it promotes the synthesis of angiogenic factors and cytokines that facilitate embryo implantation, vascularization, and immune modulation at the maternal–fetal interface [56]. Immunomodulatory HLA-G molecules in pancreatic islets are known to downregulate immune responses [57]. The *HLA-G* 14-bp indel polymorphism in the 3′UTR has been shown to influence mRNA stability and alternative splicing, thereby affecting the expression of both membrane-bound and soluble HLA-G isoforms. Finally, our results showed that *HLA-G* 14-bp indel polymorphic site analysis revealed that both the codominant and recessive models indicated an association between the Ins/Ins genotype and GDM. Individuals carrying the Ins/Ins genotype showed significantly lower levels of sHLA-G than individuals with the Del/Del genotype, potentially leading to immune activation and inflammatory responses at the maternal–fetal interface [58]. Our results align with a previous hypothesis, suggesting that inadequate HLA-G expression may contribute to maternal immune dysregulation and adverse pregnancy outcomes, including GDM [59]. This association may be influenced by the Ins allele of the polymorphic site in the 3′UTR of *HLA-G* 14-bp. A previous study reported that the recessive model of the *HLA-G* 14-bp Ins/Del polymorphism contributes to a decreased risk of autoimmune T1DM [60]. The Ins allele and Ins/Del genotype have been associated with a protective effect against autoimmune T1DM [61].

Functionally, sHLA-G is pivotal in inducing immune tolerance through interactions with inhibitory receptors, such as immunoglobulin-like transcript 2 (ILT2) (LILRB1) and ILT4 (LILRB2), which are expressed on NK cells, T cells, and antigen-presenting cells [62,63]. By engaging these receptors, sHLA-G suppresses the activity of cytotoxic T and NK cells, inhibits antigen presentation, and promotes the differentiation of tolerogenic regulatory T cells (Tregs) [64]. Therefore, the reduced levels of sHLA-G observed in Ins/Ins carriers may undermine these immunoregulatory pathways, potentially resulting in heightened immune activation at the maternal–fetal interface. This pro-inflammatory environment can exacerbate systemic inflammation and impaired glucose metabolism, leading to insulin resistance, thereby contributing to the pathophysiology of GDM.

This study has several limitations. First, the study was limited to a single center, introducing potential bias and highlighting the need for validation in multicenter studies across diverse populations. Second, the complex interplay between genetic and environmental factors in GDM pathogenesis remains unexplored. Third, the absence of IL-17A serum quantification prevented a direct correlation between the genetic variants and cytokine levels. Fourth, socioeconomic and lifestyle factors that significantly influence metabolic disorders were not accounted for, potentially confounding the results. Finally, the small sample size limits the statistical power and generalizability. Future studies with larger, well-matched cohorts and multiregional representations are necessary to strengthen these findings.

## 5. Conclusions

Overall, the study findings highlight the potential genetic factors associated with GDM, particularly the *IL-17A* -197G>A rs2275913 and *HLA-G* 14-bp indel polymorphisms. We identified a significant association between the homozygous AA codominant and recessive genotypes of *IL-17A* and GDM, as well as between the Ins/Ins genotype of *HLA-G* 14-bp in both the codominant and recessive models. These results underscore the need for further research to validate these associations and explore the biological pathways through which these polymorphisms may influence GDM development. Understanding these genetic predispositions may lead to improved screening, prevention, and management strategies for GDM.

Future research should aim to replicate these findings in larger, more diverse populations and explore the interplay between genetic and environmental factors in GDM. Investigating additional polymorphisms and their impacts on gene expression and immune regulation could enhance our understanding of GDM pathogenesis. Longitudinal studies examining the progression of GDM and its long-term effects on mothers and offspring are also recommended.

## Figures and Tables

**Figure 1 ijerph-22-00327-f001:**
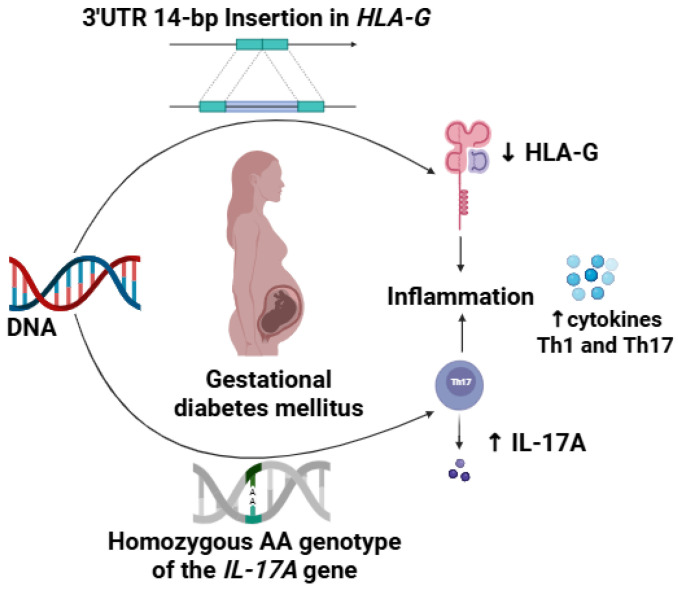
Mechanism of action of the SNPs *IL-17A* -197G>A rs2275913 and *HLA-G* 3′UTR 14-bp Insertion/Deletion in Gestational Diabetes Mellitus. AA genotype in the polymorphic loci *IL-17A* 197G>A rs2275913 causes a higher expression of IL-17, while the 14pb insertion in the 3′UTR in the *HLA-G* gene destabilizes mRNA causing a reduction in the expression of immunomodulatory HLA-G. This scenario increases the inflammatory profile characteristic of gestational diabetes mellitus with increased release of Th1/Th17 cytokines. Created in https://BioRender.com (accessed on 8 February 2025).

**Table 1 ijerph-22-00327-t001:** Primer sets and reaction conditions of polymerase chain reaction-restriction fragment length polymorphism (PCR–RFLP) experiments.

Gene	Primer Sequences (Forward, Reverse)	Cycles	Annealing Temperature (°C)	Product Size (bp)
*IL-17A*	F: AGGTACATGACACCAGAAGACC	35	60	514
	R: TGCCCACGGTCCAGAAATAC			
*IL-17RA*	F: GGAAGAGAGGAGAGGCGAAT	35	60	430
	R: CACCCCTTTGCCTGGTTCTG			
*HLA-G*	F: TGTGAAACAGCTGCCCTGTGT	30	56	345 (Del)
	R: GTCTTCCATTTATTTTGTCTCT			359 (Ins)

**Table 2 ijerph-22-00327-t002:** Current pregnancy factors and obstetric history of women with gestational diabetes mellitus and healthy control.

Variables	Group		*p* Value ^1^	Total
	GDM	Control		
N, %	79 (50.0%)	79 (50.0%)		158 (100.0%)
Current pregnancy				
Age, years	31 (26–35)	30 (25–35)	0.240	31 (25–35)
Age, >25 years	61 (77.2%)	57 (70.4%)	0.325	116 (73.4%)
Gestational age, we.	32 (27–36)	33 (24–36)	0.980	32 (25–36)
Fasting glucose, mg/dL	97 (90–106)	85 (79–90)	**0.001**	90 (82–98)
Glucose, ≥95 mg/dL	45 (57.0%)	9 (10.1%)	**0.000**	53 (33.5%)
BMI, kg/m^2^	31.3 (28–35)	28 (26–31)	**0.001**	29.5 (27.1–33.2)
Gestational BMI, *n* (%)				
Low weight	4 (5.1%)	8 (10.1%)	**0.003**	12 (7.6%)
Suitable	15 (19.0%)	23 (29.1%)		38 (24.0%)
Overweight	25 (31.6%)	34 (43.0%)		59 (37.3%)
Obesity	35 (44.3%)	14 (17.7%)		49 (31.0%)
Obesity/Overweight, *n* (%)	60 (75.9%)	48 (60.7%)	0.060	108 (68.3%)
Obesity, *n* (%)	35 (44.3%)	14 (17.7%)	**0.000**	49 (31.0%)
Obstetric background				
Children	63 (79.7%)	48 (60.7%)	**0.014**	111 (70.2%)
Abortion	29 (36.7%)	31 (39.2%)	0.870	60 (38.0%)
GDM diagnosis	6 (7.6%)	0 (0.0%)	**0.028**	8 (3.8%)
Family history of DM	59 (75.6%)	66 (84.6%)	0.228	125 (79.1%)

^1^ Bold values highlight the significant results. Abbreviations: we., weeks; BMI, body mass index; DM, diabetes mellitus; GDM, gestational diabetes mellitus.

**Table 3 ijerph-22-00327-t003:** Distribution of polymorphisms (*IL-17A* -197G>A rs2275913, *IL-17RA* -947A>G rs4819554 and HLA-G 14-bp indel).

Gene	SNP	HWE	Allele/Genotype	Group	
		0.0223031		**GDM**	**Control**
			A	38 (24.1%)	25 (15.8%)
			G	120 (75.9%)	133 (84.2%)
*IL-17A* (-197G>A)	rs2275913		G/G	50 (63.3%)	56 (70.9%)
			G/A	20 (25.3%)	21 (26.6%)
			A/A	9 (11.4%)	2 (2.5%)
		0.03841204	G	50 (31.6%)	45 (28.5%)
			A	108 (68.4%)	113 (71.5%)
*IL-17RA* (-947A>G)	rs4819554		A/A	39 (49.4%)	44 (55.7%)
			A/G	30 (38.0%)	25 (31.6%)
			G/G	10 (12.7%)	10 (12.7%)
		0.6210301	Del	93 (58.9%)	98 (62.0%)
			Ins	65 (41.1%)	60 (38.0%)
*HLA-G* 14-bp	rs66554220		Ins/Ins	15 (19.0%)	8 (10.1%)
			Ins/Del	35 (44.3%)	44 (55.7%)
			Del/Del	29 (36.7%)	27 (34.2%)

Categorical data are expressed as absolute (n) and relative (%) frequency. HWE: Hardy–Weinberg equilibrium.

**Table 4 ijerph-22-00327-t004:** Single Nucleotide Polymorphism Distribution Model Analysis.

Genetic Model	Genotype	Patients (N = 79)	Control (N = 79)	OR (95% CI)	*p* Value	FDR
***HLA-G* 14-bp indel**						
Codominant	DD	29 (36.7%)	27 (34.2%)	1	0.196	0.440
	DI	35 (44.3%)	44 (55.7%)	0.74 (0.37–1.47)	-	
	II	15 (19.0%)	8 (10.1%)	1.75 (0.64–4.77)	-	
Dominant	DD	29 (36.7%)	27 (34.2%)	1	0.739	0.8321
	DI-II	50 (63.3%)	52 (65.8%)	0.90 (0.47–1.72)	-	
Recessive	DD-II	64 (81.0%)	71 (89.9%)	1	0.112	0.335
	II	15 (19.0%)	8 (10.1%)	2.08 (0.83–5.23)	-	
***IL-17A* -197G>A rs2275913**						
Codominant	GG	50 (63.3%)	56 (70.9%)	1	0.075	0.335
	GA	20 (25.3%)	21 (26.6%)	1.07 (0.52–2.19)	-	
	AA	9 (11.4%)	2 (2.5%)	5.04 (1.04–24.44)	-	
Dominant	GG	50 (63.3%)	56 (70.9%)	1	0.310	0.557
	GA-AA	29 (36.7%)	23 (29.1%)	1.41 (0.72–2.75)	-	
Recessive	GG-GA	70 (88.6%)	77 (97.5%)	1	**0.023**	0.209
	AA	9 (11.4%)	2 (2.5%)	4.95 (1.03–23.69)	-	
***IL-17RA* -947A>G rs4819554**						
Codominant	AA	39 (49.4%)	44 (55.7%)	1	0.685	0.832
	AG	30 (38.0%)	25 (31.6%)	1.35 (0.68–2.68)	-	
	GG	10 (12.7%)	10 (12.7%)	1.13 (0.42–3.00)	-	
Dominant	AA	39 (49.4%)	44 (55.7%)	1	0.425	0.638
	AG-GG	40 (50.6%)	35 (44.3%)	1.29 (0.69–2.41)	-	
Recessive	AA-AG	69 (87.3%)	69 (87.3%)	1	1.00	1.000
	GG	10 (12.7%)	10 (12.7%)	1.00 (0.39–2.55)	-	

OR: Odds ratio; CI: Confidence interval; FDR: False Discovery Rate.

**Table 5 ijerph-22-00327-t005:** Single Nucleotide Polymorphism Distribution Model Analysis in adjustment format.

Genetic Model	Genotype	Patients (N = 78)	Control (N = 78)	OR (95% CI)	*p* Value	FDR
***HLA-G* 14-bp indel**						
Codominant	DD	29 (37.2%)	27 (34.6%)	1	**0.026**	0.059
	DI	35 (44.9%)	44 (55.1%)	0.67 (0.28–1.61)	-	
	II	14 (17.9%)	8 (10.0%)	3.34 (0.98–11.41)	-	
Dominant	DD	29 (37.2%)	27 (34.6%)	1	0.938	0.938
	DI-II	49 (62.8%)	51 (65.4%)	0.97 (0.43–2.16)	-	
Recessive	DD-II	64 (82.1%)	70 (89.7%)	1	**0.010**	**0.048**
	II	14 (17.9%)	8 (10.3%)	4.20 (1.36–12.96)	-	
***IL-17A* -197G>A rs2275913**						
Codominant	GG	50 (64.1%)	55 (70.5%)	1	**0.021**	0.059
	GA	19 (24.4%)	21 (26.9%)	1.22 (0.49–3.07)	-	
	AA	9 (11.5%)	2 (2.6%)	9.89 (1.63–59.92)	-	
Dominant	GG	50 (64.1%)	55 (70.5%)	1	0.140	0.253
	GA-AA	28 (35.9%)	23 (29.5%)	1.86 (0.81–4.30)	-	
Recessive	GG-GA	69 (88.5%)	76 (97.4%)	1	**0.006**	**0.048**
	AA	9 (11.5%)	2 (2.6%)	9.33 (1.57–55.38)	-	
***IL-17RA* -947A>G rs4819554**						
Codominant	AA	38 (48.7%)	43 (55.1%)	1	0.733	0.825
	AG	30 (38.5%)	25 (32.1%)	0.86 (0.37–2.00)	-	
	GG	10 (12.8%)	10 (12.8%)	0.62 (0.18–2.09)	-	
Dominant	AA	38 (48.7%)	43 (55.1%)	1	0.551	0.708
	AG-GG	40 (51.3%)	35 (44.09)	0.79 (0.36–1.73)	-	
Recessive	AA-AG	68 (87.2%)	68 (87.2%)	1	0.484	0.708
	GG	10 (12.8%)	10 (12.8%)	0.66 (0.21–2.09)	-	

OR: Odds ratio; CI: Confidence interval; FDR: False Discovery Rate.

## Data Availability

The datasets generated during and/or analyzed during the current study are available from the corresponding author upon reasonable request.

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
