# Peer review of "Homozygous AA Genotype of IL-17A and 14-bp Insertion Polymorphism in HLA-G 3′UTR Are Associated with Increased Risk of Gestational Diabetes Mellitus"

_ijerph, 2025, doi:10.3390/ijerph22030327_

Round 1
Reviewer 1 Report
Comments and Suggestions for Authors
Report on
“Homozygous AA genotype of IL-17A and 14-bp insertion polymorphism in the 3' UTR of HLA-G are associated with increased risk of gestational diabetes mellitus” by de Souza et al
In my opinion, this study is well planned and as per the authors first of its kind on the genetic polymorphism of IL-17 A in Brazilian population.
However, I have the follwing comments :
· The authors have not shown the time period of the subject recruitment with dates of start and the end.
· Also, the ethical approval is not clear as it doesn’t mention any date.
· The sample size is low as also pointed out by the authors. It it possible to add more subjects to the study.
· The authors should mention about the incidence of GDM in the brazilian population in the introduction with appropriate citations.
· The authors should add recent reports on The SNPs in IL-17 A (especillay in 2023 and 2024) with citations in the introduction / discussion.
· A piture/ illustration on the putative role of IL-17 A and other variables in the GDM will add more value to the article.
· The conclusion needs to be more elaborate about the study finds. The authors have added more on the future perspectives.
Author Response
Reviewer 1#
Comment 1: In my opinion, this study is well planned and as per the authors first of its kind on the genetic polymorphism of IL-17 A in Brazilian population. However, I have the following comments: The authors have not shown the time period of the subject recruitment with dates of start and the end.
Response 1: Thank you for your observation. We have added the following: “This case-control study, conducted between May 2018 and December 2019, enrolling a total of 79 pregnant Brazilian women clinically diagnosed with GDM and 79 healthy pregnant controls without comorbidities at a maternity hospital in northeastern Brazil.”(Line 95)
Comment 2: Also, the ethical approval is not clear as it doesn’t mention any date.
Response 2: We completely agree and have added more information about the ethics committee approval as requested, as follows: The study protocol was approved by the institutional ethics committee (protocol number 2.631.092 on May 2, 2018, CAAE: 73305717.2.0000.5292). (Line 105)
Comment 3: The sample size is low as also pointed out by the authors. It it possible to add more subjects to the study.
Response 3: We sincerely appreciate your suggestions. However, we must regretfully inform you that their implementation is not feasible at this time. The study was planned and executed within the constraints of limited financial resources, which precludes the possibility of genotyping new patients due to insufficient funding.
Comment 4: The authors should mention about the incidence of GDM in the Brazilian population in the introduction with appropriate citations.
Response 4: Thank you for observation. We fully agree and have added in the ‘introduction’ topic as follows: “In Brazil, the prevalence ranges from 14% to 15%”. (Line 46)
Comment 5: The authors should add recent reports on The SNPs in IL-17 A (especillay in 2023 and 2024) with citations in the introduction / discussion.
Response 5: Thank you for your valuable suggestion. We provide a recent report on the SNPs in IL-17A (ref. 35-38,40), as suggested. “Similarly, a study showed higher serum levels of IL-17 associated to A allele in rs2275913 SNP [38]. Previously, studies showed association between the rs2275913 SNP and several clinical conditions [37-40].” (Line 248)
Comment 6: A picture/ illustration on the putative role of IL-17 A and other variables in the GDM will add more value to the article.
Response 6: We appreciate the suggestion for the illustration. We have provided an illustration of the mechanism, and we truly hope it is sufficient for the aim of the study.
Comment 7: The conclusion needs to be more elaborate about the study finds. The authors have added more on the future perspectives.
Response 7: Thank you. We added in conclusion topic as follows: “We identified a significant association between the AA co-dominant and recessive genotypes of IL-17A and GDM, as well as between the Ins/Ins genotype of HLA-G 14bp in both the co-dominant and recessive models.” (Line 313)

Reviewer 2 Report
Comments and Suggestions for Authors
The study investigates the genetic polymorphisms associated with GDM in a Brazilian population, focusing on IL-17A, IL-17RA, and HLA-G genes. The findings provide valuable insights into the genetic and immunological factors potentially contributing to GDM risk, highlighting significant associations with the IL-17A -197G>A (rs2275913) and HLA-G 14bp insertion/deletion polymorphisms. The research is relevant, particularly given the increasing prevalence of GDM and the limited understanding of its genetic mechanisms. However, the manuscript would benefit from improvements in the clarity of presentation, methodological details, and contextualization of findings.
· Some of the references are not current. Please cite currently published articles (i.e. Abdominal subcutaneous fat thickness combined with a 50-g glucose challenge test at 24-28 weeks of pregnancy in predicting gestational diabetes mellitus. J Obstet Gynaecol. 2024 Dec;44(1):2329880. doi: 10.1080/01443615.2024.2329880; The reduced serum concentrations of β-arrestin-1 and β-arrestin-2 in pregnancies complicated with gestational diabetes mellitus. J Matern Fetal Neonatal Med. 2022 Dec;35(25):10017-10024. doi: 10.1080/14767058.2022.2083495).
· Details about the optimization of PCR conditions, such as annealing temperatures and cycle numbers, are missing. These are crucial for reproducibility. Include information on PCR optimization and steps taken to validate genotyping accuracy.
· The study does not mention whether genotyping was performed in duplicate or whether random samples were re-genotyped to ensure accuracy.
· The potential for contamination during PCR or gel electrophoresis is not addressed. Measures to prevent this (e.g., use of negative controls) should be detailed. Describe contamination control measures, such as the use of negative controls.
· The justification for using specific statistical tests (e.g., Mann-Whitney U test, Chi-square test) should be provided. Were these chosen based on the distribution of the data or other considerations?
· Include a power analysis to justify the sample size.
Author Response
Reviewer 2#
Comment 1: The study investigates the genetic polymorphisms associated with GDM in a Brazilian population, focusing on IL-17A, IL-17RA, and HLA-G genes. The findings provide valuable insights into the genetic and immunological factors potentially contributing to GDM risk, highlighting significant associations with the IL-17A -197G>A (rs2275913) and HLA-G 14bp insertion/deletion polymorphisms. The research is relevant, particularly given the increasing prevalence of GDM and the limited understanding of its genetic mechanisms. However, the manuscript would benefit from improvements in the clarity of presentation, methodological details, and contextualization of findings.
Some of the references are not current. Please cite currently published articles (i.e. Abdominal subcutaneous fat thickness combined with a 50-g glucose challenge test at 24-28 weeks of pregnancy in predicting gestational diabetes mellitus. J Obstet Gynaecol. 2024 Dec;44(1):2329880. doi: 10.1080/01443615.2024.2329880; The reduced serum concentrations of β-arrestin-1 and β-arrestin-2 in pregnancies complicated with gestational diabetes mellitus. J Matern Fetal Neonatal Med. 2022 Dec;35(25):10017-10024. doi: 10.1080/14767058.2022.2083495).
Response 1: We fully appreciate the suggestions. Both references were carefully added in the introduction topic as follows: “In addition to genetic and immunological factors, recent studies have indicated that clinical and anthropometric markers may improve the prediction of gestational diabetes mellitus (GDM). For example, the integration of abdominal subcutaneous fat thickness (ASFT) with the 50-g glucose challenge test (GCT) has been proposed as a potential predictive tool for GDM, emphasizing the multifactorial nature of this condition and the necessity for integrated diagnostic approaches [9]. Moreover, the predictive value of inflammatory markers, such as C-reactive protein (CRP) and interleukin-6 (IL-6), has been demonstrated to correlate with GDM risk, thereby underscoring the significance of systemic inflammation in its pathogenesis [10].” (Line 54)
Comment 2: Details about the optimization of PCR conditions, such as annealing temperatures and cycle numbers, are missing. These are crucial for reproducibility. Include information on PCR optimization and steps taken to validate genotyping accuracy.
Response 2: Thank you for observation. However, the PCR information is already provided in Table 1.
Comment 3: The study does not mention whether genotyping was performed in duplicate or whether random samples were re-genotyped to ensure accuracy.
Response 3: Thank you for your observation. The quality control was performed by re-genotyping 10% of the samples. (Line 121)
Comment 4: The potential for contamination during PCR or gel electrophoresis is not addressed. Measures to prevent this (e.g., use of negative controls) should be detailed. Describe contamination control measures, such as the use of negative controls.
Response 4: Thank you for observation. We added the following: “To minimize contamination, all PCR procedures were conducted in a designated area with distinct workspaces for DNA extraction, reaction setup, amplification, and post-PCR analysis. Aerosol-resistant tips and DNA-free reagents were utilized to prevent cross-contamination. Negative controls (reactions devoid of DNA template) were incorporated into each PCR reaction. Furthermore, pipettes were regularly decontaminated, and reagent aliquots were prepared in small volumes to mitigate the effects of repeated freeze-thaw cycles.” (Line 136)
Comment 5: The justification for using specific statistical tests (e.g., Mann-Whitney U test, Chi-square test) should be provided. Were these chosen based on the distribution of the data or other considerations?
Response 5: Thank you for this point. We have revised the methodology section to clarify this aspect (line 125). To determine the distribution of continuous variables within the study groups (case vs. control), we utilized the Shapiro-Wilk test. For variables that exhibited a non-normal distribution in at least one of the groups, we reported the median along with the 25th and 75th percentiles in the descriptive analysis. The Mann-Whitney test was employed to compare continuous variables between the groups, given their lack of normal distribution. Additionally, the Chi-square test was implemented to assess associations between categorical variables; in cases where the expected frequency in any cell of the contingency table was less than 5, we applied Fisher's exact test.
Comment 6: Include a power analysis to justify the sample size.
Response 6: Thank you for this suggestion. We performed a post hoc power analysis, and the results can be found in the revised manuscript:” A post hoc power analysis was conducted utilizing the pwr package in R based on genotype proportions in the patient and control groups. The power for IL-17A -197G>A and HLA-G 14bp indel (recessive models) was calculated as 0.63 and 0.28, respectively, which should be taken into consideration when interpreting the results.” (Line 163).

Reviewer 3 Report
Comments and Suggestions for Authors
Dear Authors,
Thank you for your article. I find the topic highly interesting and relevant; however, there are several changes that need to be addressed to enhance the quality of the manuscript.
Introduction
I have no specific recommendations for this section.
Materials and Methods
Did you include prior gestational diabetes as part of your exclusion criteria? If not, please clarify.
Results
There are two women with GDM in your control group—why were they not excluded? I believe they should be, as their inclusion might impact the validity of your findings. Additionally, how did you adjust the results shown in Table 5? You mentioned excluding one woman from your test group and two from your control group; on what basis were these exclusions made? This requires detailed clarification, as the lack of transparency raises concerns that the exclusions may have been made solely to achieve statistical significance.
Discussion
The discussion section would benefit from a more comprehensive comparison of your results with other published data. Furthermore, lines 202–223 contain highly specific details that might be difficult for a general audience to follow. I recommend replacing some of this detail with a discussion of the potential mechanisms of action, particularly focusing on the role of IL-17A in the etiology of GDM.
Your discussion of limitations needs significant expansion. For instance, one critical limitation is the absence of additional testing to measure IL-17A levels in the serum of both groups. Such data would have substantially strengthened your findings.
Conclusions
I have concerns regarding the statistical significance of the HLA-G 14 bp del/ins finding, as it appears this result was achieved after an unclear adjustment. This undermines confidence in the robustness of the conclusion and should be revisited.
I also recommend adding a figure to your manuscript, for example illustrating the mechanisms of action of the studied polymorphic markers.
Author Response
Reviewer 3#
Comment 1: Dear Authors,
Thank you for your article. I find the topic highly interesting and relevant; however, there are several changes that need to be addressed to enhance the quality of the manuscript.
Introduction
I have no specific recommendations for this section.
Materials and Methods
Did you include prior gestational diabetes as part of your exclusion criteria? If not, please clarify.
Response 1: We greatly appreciate your careful consideration. We confirm that a prior diagnosis of diabetes mellitus was indeed an important exclusion criterion in our study, as it could act as a confounding factor in participant selection. This criterion was previously reported in the methodology topic as follows: "Exclusion criteria included a history of autoimmune diseases, hypertension, prior diabetes mellitus diagnosis (fasting glucose: 96–120 mg/dL), infections, and age < 18 years."
Comment 2:
Results
There are two women with GDM in your control group—why were they not excluded? I believe they should be, as their inclusion might impact the validity of your findings. Additionally, how did you adjust the results shown in Table 5? You mentioned excluding one woman from your test group and two from your control group; on what basis were these exclusions made? This requires detailed clarification, as the lack of transparency raises concerns that the exclusions may have been made solely to achieve statistical significance.
Response 2: The authors appreciate the reviewer's observation regarding the inclusion of two women with a history of gestational diabetes in the control group. We concur that their presence may compromise the validity of the results; consequently, these participants have been excluded in the revised version of the manuscript. With respect to Table 5, adjustments were made for age, fasting blood glucose levels (greater than 95 mg/dL or less than or equal to 95 mg/dL), and family history of diabetes mellitus. The aforementioned exclusions were necessitated by a lack of information on at least one of these variables, which precluded the inclusion of these samples in the analysis conducted using the SNPassoc package in R. This package utilizes a logistic regression model to assess genetic associations and automatically excludes individuals with missing data to mitigate bias in the analysis. As outlined in the SNPassoc documentation (González et al., 2007), the package does not perform imputation of missing data and functions solely with complete observations. This information has been further elaborated in the methodology section (beginning on line 125), as recommended by the reviewer.
Comment 3:
Discussion
The discussion section would benefit from a more comprehensive comparison of your results with other published data. Furthermore, lines 202–223 contain highly specific details that might be difficult for a general audience to follow. I recommend replacing some of this detail with a discussion of the potential mechanisms of action, particularly focusing on the role of IL-17A in the etiology of GDM.
Response 3: We welcome your comments on the discussion of the manuscript and the reviewer can see that new references have been added, a new paragraph from line 292 onwards contributes to improving the quality of the discussion and the limitations of the study have been revised as recommended by the reviewers.
Comment 4: Your discussion of limitations needs significant expansion. For instance, one critical limitation is the absence of additional testing to measure IL-17A levels in the serum of both groups. Such data would have substantially strengthened your findings.
Response 4: We welcome your comments on the discussion of the manuscript and the reviewer can see that new references have been added, a new paragraph from line 292 onwards contributes to improving the quality of the discussion and the limitations of the study have been revised as recommended by the reviewers.
Comment 5:
Conclusions
I have concerns regarding the statistical significance of the HLA-G 14 bp del/ins finding, as it appears this result was achieved after an unclear adjustment. This undermines confidence in the robustness of the conclusion and should be revisited.
Response 5: Thank you for your observation. In response to the concern regarding the statistical significance of the HLA-G 14 bp del/ins finding, we would like to reiterate that participant exclusions due to missing data adhered to the SNPassoc methodology to ensure the integrity of the statistical analysis. Furthermore, we emphasize that the statistical adjustments were conducted as previously described, and the results remain consistent following these exclusions.
Comment 6: I also recommend adding a figure to your manuscript, for example illustrating the mechanisms of action of the studied polymorphic markers.
Response 6: We appreciate the suggestion for the illustration. We have provided an illustration of the mechanism, and we truly hope it is sufficient for the aim of the study.

Reviewer 4 Report
Comments and Suggestions for Authors
Dear Authors, I appreciate the opportunity to review your manuscript entitled "The Association of the IL-17A -197G>A and HLA-G 14-bp Indel Polymorphisms with GDM." The contribution is timely, considering that only a few articles have addressed this particular issue and yours pertains to a very specific population. Below, I present my comments in detail, which might be helpful for improving the scientific strength, clarity, and presentation of your work.
Major Revisions
1.0 Introduction and Background Context
1.1 Citations to Enrich the Introduction, if Desired To enhance the scientific introduction depth, especially for lines 49–51. The introduction comprehensively discusses the immunological function of HLA-G in maternal-fetal tolerance and its implications in pregnancy complications, including gestational disorders. Citation of this work will add critical insights into the role of HLA-G as a mediator of immune balance during pregnancy and its potential disruption in GDM. DOI: 10.4103/JLP.JLP_144_17 and https://doi.org/10.1159/000536559. In this introduction, the immunological role of HLA-G in maternal-fetal tolerance and its implications for pregnancy complications will be widely discussed, including gestational disorders. Citation of this work will add critical insights into the role of HLA-G as a mediator of immune balance during pregnancy and its potential disruption in GDM.
2.0 Study Design and Population
2.1 These are reasonably well-described study populations, and the case-control methodology, though well elaborated, could do with more details regarding the process of recruitment.
For example: - 2.1.1 Were participants matched for other such confounding variables like socioeconomic status or lifestyle factors?
2.1.2 Controls were screened for the following diseases, which can be related to inflammation: obesity and infections
2.2 Provide sample size justification by appropriately conducting a power calculation in order to demonstrate that the sample size is sufficient to estimate the associations under study.
3.0 Techniques of Genotyping
3.1 The Methods section describes how the genotyping was performed but most of the main technical details have not been specified:
- 3.1.1 Is genotyping by replicates or controls validated?
- 3.1.2 A priori indicate the error rate or quality control devised during all the PCR-RFLP and gel electrophoresis.
3.2 It would be necessary to give more details in regard to the selection of the SNPs and previously reported functional implications of these SNPs in GDM and related conditions.
4.0 Statistical Analysis
4.1. While the statistical tests applied were appropriate, corrections for multiple comparisons are not mentioned. Since several models were tested, a Bonferroni or false discovery rate correction is warranted and needs to be stated. This is very important, to avoid Type I errors.
5.0 Results Presentation
5.1. While the tables and figures have relevant data in them, results could be better woven into the manuscript discussion. Example:
- 5.1.1 Summarize the clinical implications of each observed ORs, like the AA genotype in IL-17A and GDM risk, in the context of overall management for GDM.
- 5.1.2 Comment on inconsistencies with other, previously published studies, investigating the same polymorphisms, also speculate on the reasons, considering population-specific factors.
5.2 These results are rather suggested to flow better: demographics first, distribution of alleles, genotypes afterward, and finally how the statistical inferences of this model are deduced.
6.0 Discussion
6.1 While not devoid of important findings, this discussion is very repetitive and lacks further elucidation of mechanisms.
Consider the following:
- 6.1.1 Further discuss the possible functional consequence of the identified polymorphisms.
- 6.1.2 Discussed implications of HLA-G 14-bp polymorphisms related to maternal-fetal immune tolerance. 6.2 Limitations of this review, particularly single-centered design bias, are that an appropriate study among other populations needs to be implemented. -
I recommend the manuscript for major revision of the various concerns outlined above, in particular the above-mentioned statistical rigors and embedding the results of the current study into previous literature. Only that will bring much bigger impacts and credibility to the manuscript. Best regards,
Author Response
Reviewer 4#
Comment 1: Dear Authors, I appreciate the opportunity to review your manuscript entitled "The Association of the IL-17A -197G>A and HLA-G 14-bp Indel Polymorphisms with GDM." The contribution is timely, considering that only a few articles have addressed this particular issue and yours pertains to a very specific population. Below, I present my comments in detail, which might be helpful for improving the scientific strength, clarity, and presentation of your work.
Major Revisions
1.0 Introduction and Background Context
1.1 Citations to Enrich the Introduction, if Desired To enhance the scientific introduction depth, especially for lines 49–51.
Response 1: Thank you for your valuable suggestions. We fully agree with the recommendations provided. In response, we have incorporated two recent references to support (references 5 and 6: 5. Terzieva, A.; Alexandrova, M.; Manchorova, D.; Slavov, S.; Djerov, L.; Dimova, T. HLA-G Expression/Secretion and T-Cell Cytotoxicity in Missed Abortion in Comparison to Normal Pregnancy. Int J Mol Sci 2024, 25, 2643; 6.Barbaro, G.; Inversetti, A.; Cristodoro, M.; Ticconi, C.; Scambia, G.; Di Simone, N. HLA-G and Recurrent Pregnancy Loss. Int J Mol Sci 2023, 24, 2557.). We truly hope these additions sufficiently substantiate the statements made in the manuscript.
Comment 2: The introduction comprehensively discusses the immunological function of HLA-G in maternal-fetal tolerance and its implications in pregnancy complications, including gestational disorders. Citation of this work will add critical insights into the role of HLA-G as a mediator of immune balance during pregnancy and its potential disruption in GDM. DOI: 10.4103/JLP.JLP_144_17 and https://doi.org/10.1159/000536559.
In this introduction, the immunological role of HLA-G in maternal-fetal tolerance and its implications for pregnancy complications will be widely discussed, including gestational disorders. Citation of this work will add critical insights into the role of HLA-G as a mediator of immune balance during pregnancy and its potential disruption in GDM.
Response 2: Thank for the suggestion. We added the following: “The immunomodulatory role of HLA-G maintains immune homeostasis during pregnancy, preventing excessive maternal immune responses against the semi-allogenic fetus. Dysregulation of HLA-G expression has been implicated in pregnancy complications, including preeclampsia and recurrent pregnant loss”
Comment 3:
2.0 Study Design and Population
2.1 These are reasonably well-described study populations, and the case-control methodology, though well elaborated, could do with more details regarding the process of recruitment.
For example: - 2.1.1 Were participants matched for other such confounding variables like socioeconomic status or lifestyle factors?
Response 3: We fully agree. However, the participants were not matched for socioeconomic status or lifestyle factors. However, potential confounding effects of these variables were acknowledged as study limitations. We added in the paragraph of the limitations as follows: “Fourth, the lack of matching for socioeconomic status and lifestyle factors.”
Comment 4: 2.1.2 Controls were screened for the following diseases, which can be related to inflammation: obesity and infections
Response 4: Thank you. The controls were not screened for obesity; however, infections were considered an exclusion criterion in our study. Therefore, we added as follows: “Exclusion criteria included a history of autoimmune diseases, hypertension, prior diabetes mellitus diagnosis (fasting glucose: 96–120 mg/dL), infections, and age < 18 years.”
Comment 5: 2.2 Provide sample size justification by appropriately conducting a power calculation in order to demonstrate that the sample size is sufficient to estimate the associations under study.
Response 5: Thank you for this suggestion. We performed a power analysis, and the results can be found in the revised manuscript.
Comment 6: 3.0 Techniques of Genotyping
3.1 The Methods section describes how the genotyping was performed but most of the main technical details have not been specified:
- 3.1.1 Is genotyping by replicates or controls validated?
Response 6: Thank you for the observation. We confirm that genotyping has been validated using DNA ladders. DNA ladders serve as molecular weight markers that help verify the sizes of the amplified DNA fragments, providing a reference for assessing the accuracy of the genotyping process. Therefore, we added as follows: “along with a standard DNA ladder” in both genotyping methods.
Comment 7: - 3.1.2 A priori indicate the error rate or quality control devised during all the PCR-RFLP and gel electrophoresis.
Response 7: Thank you for your observation. However, we did not establish a method for measuring the error rate. Prior to commencing the experiments, we conducted tests with preliminary samples.
Comment 8: 3.2 It would be necessary to give more details in regard to the selection of the SNPs and previously reported functional implications of these SNPs in GDM and related conditions.
Response 8: Thank you for your comment, but we believe that this is already covered by the text from line 71 in the introduction: "A polymorphic site in the 3′ untranslated region (UTR) of the HLA-G gene, characterized by an insertion or deletion of 14 base pairs (bp), influences mRNA stability and subsequently HLA-G protein expression [16]. Low levels of soluble HLA-G (sHLA-G) in the blood have been linked to an increased risk of gestational complications, including GDM [17,18]. Paradoxically, one study identified an association between the HLA-G del/del genotype, elevated sHLA-G concentrations, and GDM in Italian women [19]. These find-ings suggest an inconsistency in the role of HLA-G and its insertion/deletion (indel) pol-ymorphism in the pathogenesis of GDM.", that previously reported functional implications of these SNPs in GDM .
Comment 9: 4.0 Statistical Analysis
4.1. While the statistical tests applied were appropriate, corrections for multiple comparisons are not mentioned. Since several models were tested, a Bonferroni or false discovery rate correction is warranted and needs to be stated. This is very important, to avoid Type I errors.
Response 9: We appreciate your observation. To minimize the risk of type I error, we performed correction for multiple comparisons using the Benjamini-Hochberg False Discovery Rate (FDR) method. The analyses were updated, and the adjusted values were included in the revised manuscript.
Comment 10: 5.0 Results Presentation
5.1. While the tables and figures have relevant data in them, results could be better woven into the manuscript discussion. Example:
- 5.1.1 Summarize the clinical implications of each observed ORs, like the AA genotype in IL-17A and GDM risk, in the context of overall management for GDM.
Response 10: Thank you for your comment, but we believe that this is already covered in the section of the discussion that begins on line 256: “In our study, for the first time, the AA codominant and recessive genotypes of the IL-17A -197G>A rs2275913 SNP were associated with GDM. Therefore, our results support the hypothesis that the A allele acts as a probable positive regulator of IL-17A transcription and strongly contributes to the pathogenesis of GDM.”
Comment 11: - 5.1.2 Comment on inconsistencies with other, previously published studies, investigating the same polymorphisms, also speculate on the reasons, considering population-specific factors.
Response 11: Thank you for your comment, but we believe that this is already covered in the section of the discussion that begins on line 252: “A recent study found that, while the genotype frequencies of the IL-17A +45G>A rs3819025 SNP were significantly different between GDM patients and controls, with a predominance of the A allele, the -197G>A rs2275913 SNP showed no significant difference.” With regard to speculating reasons, in the passage starting at line 270 :” Interestingly, we observed a higher frequency of the A allele (68,4%) in women with GDM. We propose that there is a significant increase in the expression of IL-17RA, which supports the hypothesis that elevated IL-17A pro-inflammatory activity plays a significant role in the pathogenesis of GDM.” , and with the new text starting on line 299: “This pro-inflammatory environment could exacerbate systemic inflammation and insulin resistance, thereby contributing to the pathophysiology of gestational diabetes mellitus (GDM).”, it is clear that we are talking about the population of pregnant women.
Comment 12: 5.2 These results are rather suggested to flow better: demographics first, distribution of alleles, genotypes afterward, and finally how the statistical inferences of this model are deduced.
Response 12: Thank you for your comment and we believe that the results section meets what was suggested.
Comment 13: 6.0 Discussion
6.1 While not devoid of important findings, this discussion is very repetitive and lacks further elucidation of mechanisms.
Consider the following:
- 6.1.1 Further discuss the possible functional consequence of the identified polymorphisms.
Response 13: We appreciate the suggestion. However, we would like to clarify that we have already discussed the role of NFAT in IL-17A transcription regulation in our manuscript. Specifically, we highlighted that the rs2275913 SNP modulates gene transcription by enhancing the binding affinity of the A allele to the nuclear factor of activated T cells (NFAT), thereby promoting IL-17A expression. This mechanism is detailed in the following passage from our discussion: “The rs2275913 SNP has been proposed as a critical modulator of gene transcription, with evidence suggesting that the A allele exhibits strong affinity for the nuclear factor of activated T cells (NFAT) [34,35]. Within T cells, NFAT orchestrates regulatory activity and influences proliferation and differentiation. In conjunction with ROR-γt, activator protein 1 (AP-1), signal transducer and activator of transcription 3 (STAT3), aryl hydrocarbon receptor (AhR), interferon regulatory factor 4 (IRF4), and Runx1, NFAT binds to the IL-17 promoter region, thereby stimulating gene transcription [36]. Liu et al. concluded that the presence of the A allele is associated with the upregulation of IL-17A owing to increased affinity for NFAT [37]. Similarly, a study showed higher serum levels of IL-17 associated to A allele in rs2275913 SNP [38].”
Additionally, we added the following: “Functionally, sHLA-G plays a critical role in inducing immune tolerance by interacting with inhibitory receptors, such as ILT2 (LILRB1) and ILT4 (LILRB2), expressed on NK cells, T cells, and antigen-presenting cells [52,53]. By engaging these receptors, sHLA-G suppresses cytotoxic T-cell and NK-cell activity, inhibits antigen presentation, and pro-motes the differentiation of tolerogenic regulatory T cells (Tregs) [51]. Reduced sHLA-G levels in Ins/Ins carriers may compromise these immunoregulatory pathways, potentially leading to increased immune activation at the maternal-fetal interface. This pro-inflammatory environment may exacerbate systemic inflammation and insulin resistance, contributing to GDM pathophysiology.”
Comment 14: - 6.1.2 Discussed implications of HLA-G 14-bp polymorphisms related to maternal-fetal immune tolerance.
Response 14: Thank you. We fully agree and added the following: “In contrast, HLA-G is critical in maternal-fetal immune tolerance, as it promotes the synthesis of angiogenic factors and cytokines that facilitate embryo implantation, vascularization, and immune modulation at the maternal-fetal interface [46]. Immunomodulatory HLA-G molecules in pancreatic islets are known to downregulate immune responses [47]. The HLA-G 14-bp indel polymorphism in the 3'-UTR has been shown to influence mRNA stability and alternative splicing, thereby affecting the expression of both membrane-bound and soluble HLA-G isoforms. Finally, our results showed that HLA-G 14 bp indel polymorphic site analysis revealed that both the codominant and recessive models indicated an association between the Ins/Ins genotype and GDM. Individuals carrying the Ins/Ins genotype showed significantly lower levels of sHLA-G than individuals with the Del/Del genotype, potentially leading to immune activation and inflammatory responses at the maternal-fetal interface [48]. Our results align with a previous hypothesis, suggesting that inadequate HLA-G expression may contribute to maternal immune dysregulation and adverse pregnancy outcomes, including GDM [49].” (Line 274)
Comment 15: 6.2 Limitations of this review, particularly single-centered design bias, are that an appropriate study among other populations needs to be implemented. –
Response 15: The paragraph on limitations has been reworded as follows from line 302: “Several limitations of the study should be acknowledged. First, the sample collection was restricted to the northeastern region and primarily conducted at a single hospital, which introduces the potential for single-center design bias. Secondly, the interplay between multiple genes and environmental factors in GDM pathogenesis cannot be ruled out. Third, there was an absence of testing to quantify IL-17A levels in the serum of both groups. Fourth, the study lacked matching for socioeconomic status and lifestyle factors. Finally, the sample size may not be sufficiently large, weakening the generalizability of the findings.”

Round 2
Reviewer 2 Report
Comments and Suggestions for Authors
Thank you for the revisions
Author Response
Comment 1: Thank you for the revisions.
Response 1:
We would like to express our sincere gratitude for your thorough and thoughtful review. Your observations and suggestions have been immensely valuable in enhancing the quality of the work, and we truly appreciate the time and effort you dedicated to this process.
Reviewer 3 Report
Comments and Suggestions for Authors
Dear Authors,
Thank you for revising your article based on my recommendations. I appreciate your efforts and wish you the best in your future work!
Author Response
Comment 1: Thank you for revising your article based on my recommendations. I appreciate your efforts and wish you the best in your future work!
Response 1: We would like to express our sincere gratitude for your thorough and thoughtful review. Your observations and suggestions have been immensely valuable in enhancing the quality of the work, and we truly appreciate the time and effort you dedicated to this process.
Reviewer 4 Report
Comments and Suggestions for Authors
Dear Authors, I am appreciative for having an opportunity to review your updated manuscript. Despite improvements gained, a number of important parts of early recommendations have not yet been addressed. In the following, I present the overlooked revisions that will have to be incorporated in an effort to make your study scientifically sound, readable, and meaningful.
1. Introduction and Background Context 1.1 Missing References for Immunological Function of HLA-G The introduction doesn't have the proposed references (DOI: 10.4103/JLP.JLP_144_17 and https://doi.org/10.1159/000536559). Having these references will go a long way in enriching discussion about HLA-G's function in immunologic tolerance between mother and baby and pregnancy complications including GDM. Add these references and have a discussion about HLA-G dysregulation's immunologic processes in GDM below
3. Genotyping Techniques 3.1 Validation of Genotyping and Error Control There is no mention of replicate samples and controls for accuracy checking of genotyping. Add a statement about whether a subset of samples have been re-run for genotyping. Controls for PCR-RFLP and electrophoresis: There is no mention of an expected level of errors, and no supplementary controls for minimizing errors in genotyping. All such information is relevant in ensuring accuracy in the information. 3.2 Selection of SNPs and Functional Consequences The paper doesn't state explicitly why these SNPs have been selected apart from citing studies in the past. There must have been a strong reason, particularly in terms of their functional contribution in GDM.
5. Results Presentation 5.1 Clinical Interpretation of Measured Odds Ratios (ORs The debate neglects to put into its rightful position the clinical significance of ORs. For instance, in patient care, as it pertains to susceptibility to GDM, what is the effect of AA genotype of IL-17A? Comparison with work previously published is lacking. There is a need for increased discussion regarding why specific findings agree with, and/or vary from, work previously published and in what manner population-specific genetic factors could cause variation.
6. Discussion 6.1 Functional Consequences of Polymorphisms Detected The potential for a biological impact of IL-17A and HLA-G polymorphisms in GDM pathogenesis is not addressed in detail enough. Describe in detail how such variants can affect cytokine regulation, immune tolerance, and insulin resistance. 6.2 HLA-G and Maternal-Fetal Immune Tolerance Although HLA-G is referred to, its role in the processes of maternal-fetal tolerance is not clearly explained sufficiently. Describing this would strengthen the mechanism of the findings. 6.3 Limitations and Generalizability Although some have been discussed, significant ones remain unexplored: The single-center bias and requirements for replication in numerous populations are not emphasized. The incompatibility of socioeconomic and life remains a problem.
Overhaul in its Entirety The manuscript yields fascinating results, but a few key methodological and interpretative concerns remain underdeveloped. These must be resolved before the study can become acceptable for publication. I urge the authors to finish including any missing revisions and add a discussion with increased mechanistic thinking.
Author Response
Comment 1:
Dear Authors, I am appreciative for having an opportunity to review your updated manuscript. Despite improvements gained, a number of important parts of early recommendations have not yet been addressed. In the following, I present the overlooked revisions that will have to be incorporated in an effort to make your study scientifically sound, readable, and meaningful.
- Introduction and Background Context
1.1 Missing References for Immunological Function of HLA-G
The introduction doesn't have the proposed references (DOI: 10.4103/JLP.JLP_144_17 and https://doi.org/10.1159/000536559). Having these references will go a long way in enriching discussion about HLA-G's function in immunologic tolerance between mother and baby and pregnancy complications including GDM. Add these references and have a discussion about HLA-G dysregulation's immunologic processes in GDM below
Response 1: Thank you. We agree with your suggestion and have added the recommended references [ref 19-21] to the Introduction.
A discussion about HLA-G dysregulation's immunologic processes in GDM was included “In contrast, HLA-G 14 bp indel polymorphism influences the immune response by modulating sHLA-G levels. This interplay between the pro-inflammatory and anti-inflammatory pathways at the molecular level is crucial for understanding how immune dysregulation contributes to GDM development.” (LINE 100)
Comment 2:
- Genotyping Techniques
3.1 Validation of Genotyping and Error Control
There is no mention of replicate samples and controls for accuracy checking of genotyping. Add a statement about whether a subset of samples have been re-run for genotyping. Controls for PCR-RFLP and electrophoresis: There is no mention of an expected level of errors, and no supplementary controls for minimizing errors in genotyping. All such information is relevant in ensuring accuracy in the information.
Response 2: Thank you for your comment. We already reported the re-run as follows: “The quality control was performed by re-genotyping 10% of the samples”. In addition, the expected error rate was assessed using replicate analysis.
Comment 3:
3.2 Selection of SNPs and Functional Consequences The paper doesn't state explicitly why these SNPs have been selected apart from citing studies in the past. There must have been a strong reason, particularly in terms of their functional contribution in GDM.
Response 3: Thank you. We added the following on the ‘Introduction’ topic: “Molecularly, these SNPs can affect the balance between proinflammatory and anti-inflammatory signaling. IL-17A and IL-17RA polymorphisms may increase proinflammatory cytokine production, driving an imbalance in favor of inflammation, which is a characteristic feature of GDM. In contrast, HLA-G 14 bp indel polymorphism influences the immune response by modulating sHLA-G levels. This interplay between the pro-inflammatory and anti-inflammatory pathways at the molecular level is crucial for understanding how immune dysregulation contributes to GDM development.” (LINE 97)
Comment 4:
- Results Presentation
5.1 Clinical Interpretation of Measured Odds Ratios (ORs The debate neglects to put into its rightful position the clinical significance of ORs. For instance, in patient care, as it pertains to susceptibility to GDM, what is the effect of AA genotype of IL-17A? Comparison with work previously published is lacking. There is a need for increased discussion regarding why specific findings agree with, and/or vary from, work previously published and in what manner population-specific genetic factors could cause variation.
- Discussion
6.1 Functional Consequences of Polymorphisms Detected The potential for a biological impact of IL-17A and HLA-G polymorphisms in GDM pathogenesis is not addressed in detail enough. Describe in detail how such variants can affect cytokine regulation, immune tolerance, and insulin resistance.
6.2 HLA-G and Maternal-Fetal Immune Tolerance Although HLA-G is referred to, its role in the processes of maternal-fetal tolerance is not clearly explained sufficiently. Describing this would strengthen the mechanism of the findings.
Response 4: We fully understand your concern. Accordingly, we have incorporated the following discussion:
Similarly, a study showed higher serum levels of IL-17 associated to A allele in rs2275913 SNP [44]. (LINE 274)
IL-17 is implicated in insulin resistance through multiple mechanisms, including the activation of inflammatory cells such as macrophages, which subsequently secrete pro-inflammatory cytokines, such as TNF-α and IL-6 [49]. These cytokines disrupt normal insulin signaling in target cells [50]. Moreover, IL-17 can directly interfere with the insulin signaling pathway, particularly by acting on differentiated adipocytes, leading to impaired cellular responsiveness to insulin and reduced glucose uptake [51]. (LINE 286)
In contrast, HLA-G is critical in maternal-fetal immune tolerance, as it promotes the synthesis of angiogenic factors and cytokines that facilitate embryo implantation, vascularization, and immune modulation at the maternal-fetal interface [56]. Immunomodulatory HLA-G molecules in pancreatic islets are known to downregulate immune responses [57]. The HLA-G 14-bp indel polymorphism in the 3'-UTR has been shown to influence mRNA stability and alternative splicing, thereby affecting the expression of both membrane-bound and soluble HLA-G isoforms. Finally, our results showed that HLA-G 14 bp indel polymorphic site analysis revealed that both the codominant and recessive models indicated an association between the Ins/Ins genotype and GDM. Individuals carrying the Ins/Ins genotype showed significantly lower levels of sHLA-G than individuals with the Del/Del genotype, potentially leading to immune activation and inflammatory responses at the maternal-fetal interface [58]. Our results align with a previous hypothesis, suggesting that inadequate HLA-G expression may contribute to maternal immune dysregulation and adverse pregnancy outcomes, including GDM [59]. (LINE 306)
Functionally, sHLA-G is pivotal in inducing immune tolerance through interactions with inhibitory receptors, such as immunoglobulin-like transcript 2 (ILT2) (LILRB1) and ILT4 (LILRB2), which are expressed on NK cells, T cells, and antigen-presenting cells [62,63]. By engaging these receptors, sHLA-G suppresses the activity of cytotoxic T and NK cells, inhibits antigen presentation, and promotes the differentiation of tolerogenic regulatory T cells (Tregs) [64]. Therefore, the reduced levels of sHLA-G observed in Ins/Ins carriers may undermine these immunoregulatory pathways, potentially resulting in heightened immune activation at the maternal-fetal interface. This pro-inflammatory environment can exacerbate systemic inflammation and impaired glucose metabolism, leading to insulin resistance, thereby contributing to the pathophysiology of GDM. (LINE 324)
Comment 5:
6.3 Limitations and Generalizability Although some have been discussed, significant ones remain unexplored: The single-center bias and requirements for replication in numerous populations are not emphasized. The incompatibility of socioeconomic and life remains a problem.
Response 5: Thank you. We have revised the paragraph as follows to address your suggestion: “This study has several limitations. First, the study was limited to a single center, introducing potential bias and highlighting the need for validation in multicenter studies across diverse populations. Second, the complex interplay between genetic and environmental factors in GDM pathogenesis remains unexplored. Third, the absence of IL-17A serum quantification prevented a direct correlation between the genetic variants and cytokine levels. Fourth, socioeconomic and lifestyle factors that significantly influence metabolic disorders were not accounted for, potentially confounding the results. Finally, the small sample size limits the statistical power and generalizability. Future studies with larger, well-matched cohorts and multiregional representations are necessary to strengthen these findings.” (LINE 334)